# Alloplastic Epidermal Skin Substitute in the Treatment of Burns

**DOI:** 10.3390/life14010043

**Published:** 2023-12-27

**Authors:** Aleksandra Barbachowska, Tomasz Korzeniowski, Agnieszka Surowiecka, Jerzy Strużyna

**Affiliations:** 1East Center of Burns Treatment and Reconstructive Surgery, 21-010 Lęczna, Poland; aleksandrabarbachowska@gmail.com (A.B.); dr.surowiecka@gmail.com (A.S.); jerzy.struzyna@gmail.com (J.S.); 2Department of Plastic, Reconstructive Surgery and Burn Treatment, Medical University of Lublin, 20-093 Lublin, Poland; 3Department of Plastic and Reconstructive Surgery and Microsurgery, Medical University of Lublin, 20-093 Lublin, Poland

**Keywords:** burn wound, healing, skin substitute, pain reduction, epidermal skin substitute, synthetic skin substitute

## Abstract

The goal of burn wound treatment is to ensure rapid epithelialization in superficial burns and the process of rebuilding the lost skin in deep burns. Topical treatment plays an important role. One of the innovations in the field of synthetic materials dedicated to the treatment of burns is epidermal skin substitutes. Since the introduction of Suprathel^®^, the alloplastic epidermal substitute, many research results have been published in which the authors investigated the properties and use of this substitute in the treatment of wounds of various origins, including burn wounds. Burn wounds cause both physical and psychological discomfort, which is why ensuring comfort during treatment is extremely important. Alloplastic epidermal substitute, due to its biodegradability, plasticity, no need to remove the dressing until healing, and the associated reduction in pain, is an alternative for treating burns, especially in children.

## 1. Introduction

The skin is the largest human organ that ensures homeostasis of the body. The loss of its integrity may lead to disturbances in thermoregulatory and protective function against harmful external factors such as microorganisms, radiation, or temperature changes. Damage to this barrier increases the risk of infection, water loss, hypothermia, and even death of the organism [1,2]. Such changes are also caused by burns through the destructive effects of heat, leading to various levels of skin damage. Due to the type of energy, burns can be mainly divided into thermal, chemical, electrical, radiation-induced, and mixed [1].

Damage to the skin as a result of burns can affect its various layers, including the epidermis, dermis, and subcutaneous tissue. Over the years, many classifications of burn depth have been created and introduced. In Europe, burns are most often divided into superficial, deep dermal, and full-thickness burns [1]. Superficial burns are usually red, dry, and painful. They affect only the epidermis and heal spontaneously by its regeneration within about a week. Superficial partial-thickness burns involve the superficial part of the dermis. They are usually red, moist with blisters, and may be very painful. With proper treatment, they can also heal on their own within 14 days. Deep dermal burns extend to a deeper layer the dermis, damaging the hair follicles and glandular tissue. They can take on a variety of appearances, including red and wet with blisters, but also waxy and dry. They show the ability to heal spontaneously within 2 to 9 weeks, especially after enzymatic debridement; however, early wound closure with skin grafting is often required to avoid hypertrophic scarring and contractures. Full-thickness burns with underlying subcutaneous tissue do not heal spontaneously and also require surgical treatment [3]. Early excision of deep burns has been the treatment of choice for many years. The main goal of early removal of necrotic tissue is to prevent infection. The necrotic eschar is an excellent breeding ground for microorganisms which, apart from stimulating a local inflammatory reaction and tissue infection, may end in systemic inflammatory reactions and lead to sepsis and multiple organ failure [4,5]. Advances in burn wound treatment allow for more effective removal of eschar using less invasive techniques, such as enzymatic debridement [6]. Since loss of skin due to thermal injury leads to serious multiorgan complications, burn wound treatment still remains a challenge and novel medical devices are introduced to accelerate wound healing, prevent infection, reduce pain, and improve postburn scarring. Since there are a lot of dressings on the market, our objective is to examine the epidermis substitute in more detail because it appears to be a promising alternative for treating burns.

## 2. Methods

Three independent researchers (AB, AS, TK) filtered medical databases (PubMed, PubMed Central, MEDLINE) in search of articles describing the role of Class III medical devices for epidermal replacement in burn wound healing. The inclusion filters were “burn” or “burn wound” and “Suprathel^®^” or “alloplastic skin substitute” or “alloplastic epidermal substitute” or “synthetic one-time application epidermal substitute” or “epidermal substitute”. Titles, abstracts, and full texts in English were verified to choose original articles and reviews. The search strategy identified 25 records, out of which 16 studies and articles were included in the study. Studies that were excluded did not particularly address an alloplastic epidermal replacement.

## 3. Results

### 3.1. Burn Wound Management

The goal of burn wound treatment is to ensure rapid epithelialization in superficial burns and the process of rebuilding the lost skin in deep burns. Topical treatment plays an important role in this process. It aims to create optimal conditions for spontaneous regeneration of the skin defect or preparation of wounds for transplantation after removal of necrotic tissues. Creating optimal conditions for healing involves, among other things, removing fibrin, necrotic tissues, and excessive secretions from the wound surface. In addition, it protects tissues from drying out by ensuring constant humidity [7,8]. However, dressings that are ideal for the treatment of deep burn wounds have not been developed so far, which would enable their complete healing without the need for multiple dressing changes, additional surgical interventions and the final closure of skin defects with split-thickness skin grafts. Currently, it is a standard operating procedure in the treatment of deep burns, preceded by early surgical or enzymatic removal of the necrotic tissue [9,10].

Autologous split-thickness skin grafts are the primary method for closing full-thickness skin burns. However, in the case of extensive burns involving large skin surfaces, the lack of donor sites may prolong the healing process [11]. Extending this time may adversely affect the burn, increasing the risk of infection and prolonged hospitalization. Reducing the healing time of deep burn wounds, including deep dermal and full-thickness burns, would bring many benefits. For this purpose, biological and synthetic materials dedicated to the treatment of deep burns are developed [7,12].

An optimally selected dressing for the treatment of deep dermal burns should primarily support the process of tissue regeneration, protect against infection, and reduce the patient’s discomfort during dressing changes, including minimizing pain and treatment costs [13]. In addition, each time the type of dressing should be matched to the current condition of the wound, taking into account its depth and extent. 

The wound healing process includes the following phases: hemostasis, inflammation, proliferation, and remodeling. For healing to be successful, these stages must be completed in the correct order. The initiation of the inflammatory phase is crucial because numerous immune system cells are involved in the healing of burn wounds (Figure 1) [14,15]. Moreover, in the event of dysregulation of the proliferative phase, this process slows down or even stops completely, causing the conversion of a burn wound which, without appropriate surgical treatment and coverage with skin grafts, will turn into a deep chronic wound [16].

The potential of spontaneous healing is reserved for superficial and superficial partial-thickness burns in which the dermis is not fully destroyed [17,18]. The residual stem cells present in the skin adnexa are translocated after the burn trauma and stimulate the healing process [19,20,21,22,23,24]. In the initial phases of wound healing, residual mesenchymal stem cells from the subcutaneous white fat tissue and skin adnexa secrete various growth factors and chemokines regulating the inflammation process [25,26]. They regulate the proliferation and maturation of macrophages’ populations M1 and M2 [27], regulate secretion of metalloproteinases, promote secretion of extracellular matrix proteins and glycosaminoglycans, as well as prevent dermal cells from free radicals and oxidative stress [28]. An M2 macrophage response that is well controlled is necessary for optimal wound healing. Th2 cytokines cause these to become active. The induction of resolving macrophages, which are triggered by the phagocytosis of apoptotic cells as a consequence of inflammation, is also significant. Excessive M2 macrophage activation causes tissue fibrosis and hypertrophic scarring, whereas insufficient M2 macrophage response impairs wound healing [29,30,31]. At the basal membrane, there is also an important population of residual keratinocyte stem cells, prekeratinocytes, responsible for keratinocyte maturation [19]. Spontaneous burn wound healing is dependent on the uninjured stem cells from skin adnexa and subcutaneous white fat tissue and keratinocyte precursors but also requires a moist environment, prevention from infection and secondary traumas.

Therefore, burn wounds with epithelialization potential should receive optimal conservative treatment to achieve effective healing without complications. A dressing that creates and maintains a moist environment ensures optimal conditions for wound healing. Moisture not only increases the rate of epithelial formation, but also its quality, maintaining wound exudate that contains cytokines and essential proteins in response to injury. It also improves the production of collagen by fibroblasts and supports the synthesis of the extracellular matrix [32,33]. Dry dressing does not have these properties. Additionally, removing dressing stuck to the wound may cause secondary damage to the newly formed epidermis. In dry wounds, keratinocytes migrate to a deeper level to reproduce most efficiently, but in a moist wound environment, keratinocytes can more easily migrate toward the wound surface for closure. Moreover, a humid environment may reduce the risk of infection through hypoxia. A hypoxic wound bed increases angiogenesis and reduces pH, which results in less colonization by bacteria [33,34]. For this reason, the most commonly used dressings are those that provide a moist, optimal environment in the wound, such as hydrocolloid, hydrogel, paraffin, silicone, and silver dressings [35]. However, biological and synthetic materials, such as epidermis and skin substitutes, which temporarily or permanently take over the functions of the skin are gaining in popularity (Figure 2). They have similar purposes as mentioned above; usually they are easy to handle and adapt to the surface and shape of the wound [36]. Interestingly, there are more than 70 commercially available skin substitutes, many of which are used in the treatment of burns [37].

Allografts and xenografts with amniotic membrane are temporary biological replacements for the epidermis. Xenografts have been used in the treatment of burns since the 1960s [38]. Some clinical studies indicate that these dressings also have a positive effect on the healing process, as they diminish the risk of infection and reduce the frequency of dressing changes [39]. The most recognized products of animal origin include porcine skin and porcine intestinal submucosa but also fish skin, especially the Nile tilapia skin. The last one might be as effective as previous xenografts dressings based on the treatment of experimental burns [40,41]. Cryo- and glycerol-preserved allografts can also provide good temporary wound coverage in case of the lack of autografts. Because of stimulating the recipient’s immune response and related side effects it is considered as a temporary dressing preparing the wound bed for final closure [42]. Allografts are most commonly harvested during multiorgan retrievals and availability is dependent from transplantation law and condition in a given country or region [43]. Amniotic membrane can be used on partial-thickness burns and skin donor sites, and also on freshly excised burn wounds. It also has great ability to accelerate re-epithelialization and promote angiogenesis and wound healing. On the other hand, it is a biological dressing and it carries a risk of viral infection transmission, e.g., hepatitis, syphilis, and AIDS. This important aspect limits their use [44,45]. Also, human amniotic membranes are donated by healthy volunteers and the sources might be limited [43].

Moving on to synthetic products, polymers will be discussed below. Another example of a synthetic alternative is a product known as Omiderm^®^, which shares amino acid similarities with human epidermis (Acrylamide and hydroxymenthyl methacrylate). It is a hydrophilic copolymer membrane made of poliurethane which is non adhesive, adaptable, and transparent [46]. Based on the studies, it also reduces pain, protects the growth of bacteria, and thanks to its transparency, it allows access to the wound bed without changing the dressing [47]. As they are synthetic, they do not stimulate the immune response of the recipient and do not carry the risk of transmitting viruses and other diseases.

Autologous keratinocytes might be classified as durable epidermal substitutes. Their use in the treatment of burns was first described in 1981 by N. E. O’ Connor [48]. The greatest advantage of these autologous keratinocytes seems to be the ability to cover a large burn wound obtained from a small biopsy of an uninjured part of the skin. However, due to the need to wait up to 3 weeks for culturing keratinocyte cells, the wound requires temporary coverage with another dressing [49]. Cultured autologous keratinocytes might be used as a stand-alone dressing or as an additional dressing for autologous skin grafts [50]. This method can significantly shorten the healing time [51]. When it comes to synthetic, durable substitutes for the epidermis, the ongoing challenge is to create a durable substitute for the epidermis that would heal into the wound and act as a final dressing. This would certainly be a great milestone in the treatment of burns, which would change the approach and treatment process.

### 3.2. Properties of the Epidermal Substitute

One of the latest innovations in the field of synthetic materials dedicated to the treatment of burns is Suprathel^®^, which is an alloplastic substitute for the epidermis (Figure 3).

The product has been patented by PolyMedics Innovations GmbH (Denkendorf, Germany). It is produced from a synthetic copolymer consisting mainly of DL-lactide (>70%), trimethylene carbonate, and e-caprolactone. In the production process, monomers are polymerized in the melting procedure and then dissolved in organic solvents. The resulting material is treated with a suitably modified phase inversion and lyophilization technique. The end product of these processes is a microporous membrane [12,52,53]. This structure prevents the accumulation of exudate in the wound, but at the same time creates a moist environment, which is intended to enable wound healing and epithelialization [54]. In addition, it provides high plasticity and thus immediate adjustment of the dressing to the wound at body temperature. This is especially useful in hard-to-reach locations and gives a possibility to be applied to anybody area in children as well as in adults [54]. When applying the dressing, it is recommended to cover it with one layer of paraffin gauze, another layer of absorbent gauze, and to secure the dressing against movement of the material. During wound healing, the dressing detaches from its surface as a result of a decrease in molecular weight [52].

According to the manufacturer, it is dedicated to the treatment of superficial and deep dermal burns with small full-thickness areas (3°), diseases associated with skin loss, such as donor sites and traumatic wounds. In addition, it can also be used in the treatment of epidermal abrasions, frostbite, in reconstructive surgery, correction of scars after dermabrasion, as well as skin diseases of dermatological origin, e.g., toxic epidermal necrolysis (TEN).

The exact mechanism of wound healing under the alloplastic membrane is still being discussed. There are several mechanisms in which polylactide-based copolymers might accelerate wound epithelialization. In the histopathological specimens from rat oral mucosa, local epithelial hyperplasia without serious hiperkeratosis was observed [55]. When applied on the wound, the polylactic polymers are degraded by dermal enzymes [56]. The products of degradation: lactic acid, lactate, lactide promote neovascularization, most likely by stimulating residual mesenchymal stem cells to secrete growth factors and chemokines for various cells, including endothelial cells [57]. Also, interaction with polylactates promotes secretion of proangiogenic endothelial growth factor (EDGF) from the macrophages [15] as well as transforming growth factor-β (TGF-β) [58]. When injected into the dermis, polylactide polymers induce macrophages to secrete various cytokines, like interleukin 1 (IL-1), interleukin 6 (IL-6), interleukin 8 (IL-8), and promote collagen synthesis [58].

Certain concentrations of lactates can also interfere with dermal fibroblasts to promote their proliferation and dermal healing [59]. The nanoporous structure of the membrane improves wound healing by simulating an extracellular matrix scaffold formation. The matrix is important for cellular migration and adhesion [60]. This phenomena of cellular migration might be one of the most important potential mechanisms of fast re-epithelialization in superficial and superficial partial-thickness burns. The alloplastic epidermal substitute is also water permeable, which enables a moist environment of the burn wound that promotes healing [61]. What is more, the nanoporous sheet provides oxygen ventilation [60]. A poly L-lactic acid nanosheet can also prevent the wound from bacterial wound infection [62].

The safety of polylactic biopolymers relies on the biocompatibility and biodegradability of the products. In clinics, these materials are commonly used for the production of surgical sutures and stents [63]. They can be used in pediatric patients for scalds. Figure 4 shows an example of application in a 2-year-old child after a scald.

### 3.3. Review of the Literature on Epidermal Skin Substitutes

Since the introduction of Suprathel^®^, many research results have been published in which the authors investigated the properties and use of this product in the treatment of wounds of various origins, including burn wounds (Table 1).

In 2007, Uhlig et al. demonstrated pain reduction when using Suprathel^®^ on donor sites after split-thickness skin grafts compared to standard paraffin dressings. The Visual Analogue Scale (VAS) was used to assess pain. The study was discontinued after analyzing 20 patients, as the results significantly showed that the use of alloplastic epidermal substitute is significantly less painful for patients than treatment with conventional donor sites dressings (*p* = 0.0002). In the second part of the study, the use of Suprathel^®^ and Omiderm^®^ in the treatment of superficial burn wounds was compared. In this section, Suprathel^®^ was also found to be significantly less painful compared to Omiderm^®^, a hydrophilic polyurethane film developed to treat burn wounds. In addition, investigators rarely observed rolling of Suprathel^®^ in the peripheral areas of the wound as a result of mechanical intervention during dressing changes compared to standard paraffin gauze and Omiderm^®^. Another advantage was its ability to degrade, which allowed the dressing to be separated from the epithelial wound underneath. This study enabled Suprathel^®^ to obtain approval for the use of this product on split-thickness skin graft donor sites, partial-thickness burn wounds, and burn-like lesions, such as abrasions and Lyell’s syndrome [12,13]. Madry et al. showed that Suprathel^®^ can perform a temporary function of the epidermis in burns, split-thickness frostbite, and toxic epidermal necrolysis (TEN). In addition, they demonstrated its flexibility, which allows for application in difficult locations, such as fingers and toes, and the ability to leave the product in the wound bed until complete healing. It minimizes the pain associated with changing the dressing [60]. However, they noticed that its effectiveness decreases with the delay of application and with increasing burn depth [78]. Lindford et al. concluded that Suprathel^®^ in TEN has advantages over allogeneic grafts. They also observed less wound exudate, less pain, easier application, and earlier re-epithelialization [65]. Keck et al. showed that the healing time of deep dermal burns with the use of Suprathel^®^ was longer compared to skin grafts, but the final effect of scar quality at 30 and 90 days after treatment was comparable [66]. Galati et al. studied the treatment of deep burn wounds on the hands using Suprathel^®^ and split-thickness skin grafts. They concluded that after tangential excision, both methods may be equivalent [67]. Sari et al. also described an interesting case in which they successfully used Suprathel^®^ in the treatment of hand deformities in a patient with dystrophic epidermolysis bullosa [68]. In 2011, Kaartinnen and Kuokkanen published the results of a prospective study comparing the effect of Suprathel^®^ and Mepilex^®^ Transfer on donor sites healing after split-thickness skin grafting. Significantly lower pain intensity was recorded on the first and fifth postoperative day assessed on the VAS in favor of Suprathel^®^. In addition, less bleeding and exudation was also observed in the Suprathel^®^ group compared to the Mepilex^®^ Transfer group [69]. Schwarze et. al. also evaluated the impact of Suprathel^®^ on wound healing in donor sites of split-thickness skin grafts compared to Jelonet^®^( Smith & Nephew, Watford, UK). Although there was no noticeable difference in healing time and re-epithelization, patients treated with Suprathel^®^ experienced significantly less pain and required less frequent dressing changes [52].

Schiefer et al. compared Suprathel^®^ and Epicite Hydro^®^ (QRSKIN GmbH; Würzburg, Germany) (a bionanocellulose (BNC) and 95% water dressing) for the treatment of split-thickness burns. The primary endpoints assessed in this study were infection, bleeding, exudate, pain, and dressing change. Both materials did not require changing during the study period, and no signs of infection or bleeding were observed. Also, there was no significant difference in the amount of exudate, pain intensity, and epithelialization time between the tested products [70]. Haller et. Al. compared synthetic epidermal skin substitute and porcine xenograft in the treatment of partial-thickness burns. Even though the groups differed slightly in TBSA, it showed a significantly lower necessity for skin grafts and lower infection rate in favor of Suprathel^®^ [71]. It is also worth mentioning that in in vitro studies, Suprathel^®^ showed an excellent bactericidal effect superior to that of Acticoat^®^ and Aquacel^®^ [72]. Schiefer et al. reported on the assessment of scar quality following Suprathel^®^ and Dressilk^®^ (Prevor, Valmondois, France) treatment for superficial burns in another investigation. Other than a little bit more pigmentation while using Dressilk^®^, both dressings produced results that were comparable in the subjective scar evaluation. Aside from a few minor variations in favor of Dessilk^®^, there were no notable differences in the objective scar evaluation [79].

Recently, there are more papers in the literature in which the authors describe the results of treating burn wounds in children with the use of Suprathel^®^. Children constitute an important group of burn victims and about 84% of them suffer from scald burns [80]. This type of burns can lead to the injury of the critical parts of the body including head, neck, upper, and lower limbs [80]. In 2016, Rashaan et al. published the results of their prospective study in which children with partial-thickness burns were treated. This study showed the potential benefits of Suprathel^®^ treatment in terms of pain and scar formation. However, the study group consisted of only 21 patients [54]. In a preliminary report on enzymatic debridement in children, Suprathel^®^ was used successfully for covering mixed deep dermal and full thickness burn wounds [81]. Furthermore, Weitzmann et al. conducted a comparison between Suprathel^®^ and Jelonet^®^ in the adult patient group following enzymatic debridement. Interestingly, patients who used Suprathel^®^ experienced less pain and greater comfort, and there was no discernible difference in the amount of time it took for their wounds to heal [82]. One of the most recent studies on this topic is by Schriek et al., in which they included a total of 2084 children treated for superficial and deep split-thickness skin burns. The study compared Suprathel^®^ with alternative dressings. The average number of procedures was statistically different, which was 54.35% lower in the group of patients treated with Suprathel^®^ than in the control group (*p* < 0.0001). In the study group, 91.74% of children managed to be treated conservatively, compared to the control group, in which 23.95% of patients required split-thickness skin grafts. In addition to the therapeutic outcome, fewer interventions significantly reduced treatment costs [73]. Interestingly, another study compared Suprathel^®^ to Mepilex^®^ Ag for the treatment of partial-thickness scalds in children. They reported no significant differences in any of the outcomes, but the study group included only 58 children [74]. Moreover, Miguel-Ferrero et al. used Suprathel^®^ in the treatment of TEN in a four-year-old patient whose skin lesions covered over 60% of the body surface. They observed complete re-epithelialization after 16 days and during the healing process, patients reported minimal or no pain [75]. A simple study on the use of Suprathel^®^ in the treatment of a group of children including 65 patients with superficial dermal, mid-dermal, and deep dermal burns with a median 23.6% TBSA showed an average healing time of 15 days [76]. Pain reduction is very important in the process of wound healing. Alloplastic epidermal substitute was also reported in a case of moist desquamation due to radiation. Application of polylactic copolymer reduced pain and enabled administration of the full radiation protocol preventing skin from deep radiative injury. After 10 days from application on the injured side, full re-epithelialization was observed [77].

## 4. Conclusions

Local treatment of burns is a key element of therapy. Burn wounds cause both physical and psychological discomfort, which is why ensuring comfort during treatment is extremely important. Epidermal skin substitute may become a very good option due to its biodegradability, plasticity, no need to remove the dressing until healing, and the associated reduction in pain, especially in children. These materials are a therapeutic option in superficial and superficial partial-thickness burns. Further clinical trials that are randomized and conducted across multiple centers are necessary to confirm the long-term effects and precise influence on scar development.

## Figures and Tables

**Figure 1 life-14-00043-f001:**
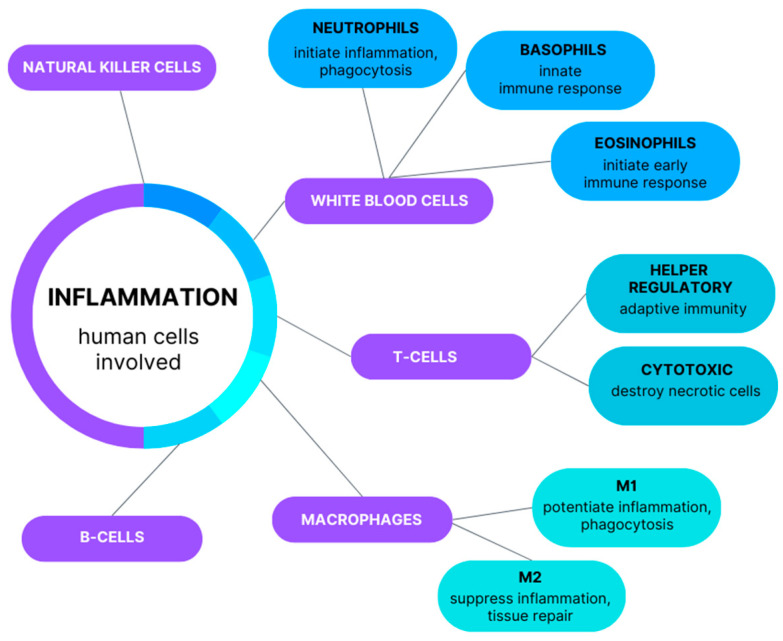
Immune response cells in the healing of burn wounds.

**Figure 2 life-14-00043-f002:**
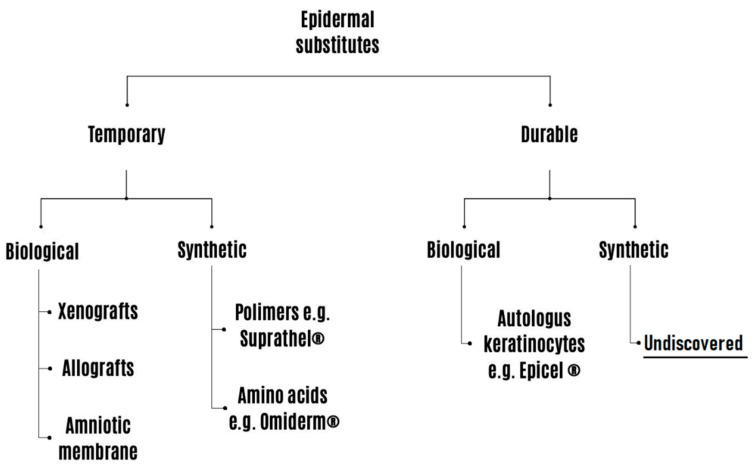
Epidermal substitutes classification.

**Figure 3 life-14-00043-f003:**
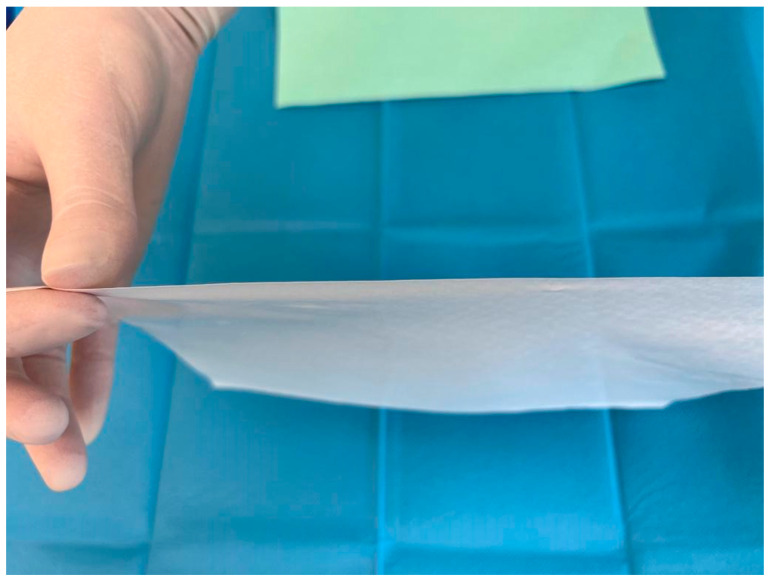
Alloplastic epidermal substitute.

**Figure 4 life-14-00043-f004:**
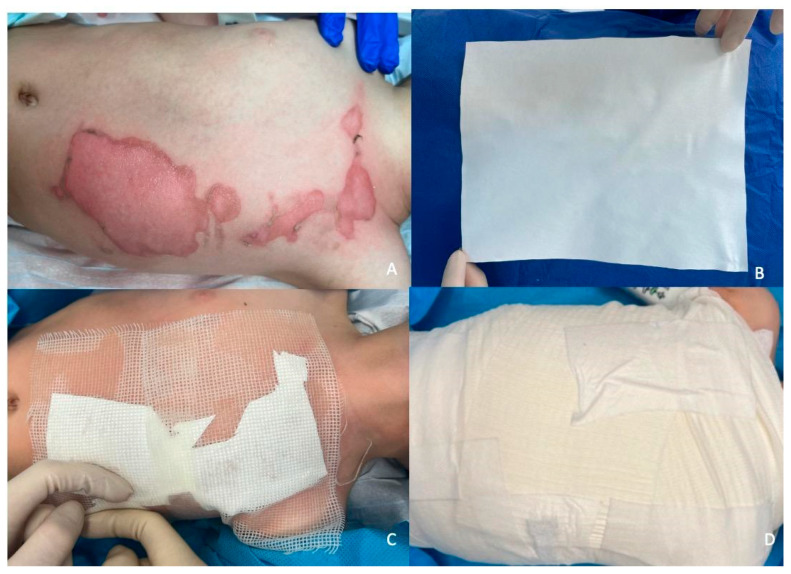
Utilizing of Suprathel^®^ (PolyMedics Innovations GmbH, Germany). (**A**) Scald after surgical debridement. (**B**) Suprathel^®^ prior to use. (**C**) Suprathel^®^ layered in a paraffin-coated cotton dressing. (**D**) External bandage and gauze dressing.

**Table 1 life-14-00043-t001:** Summary of articles on alloplastic epidermal substitute.

Study	Study Type	Dressing Activity	Patients and Methods	Outcomes
Suprathel^®^-an innovative, resorbable skin substitute for the treatment of burn victims [12]	Randomized	Reducing pain symptoms	Comparison of Suprathel^®^ with paraffin gauze applied to the donor fields after harvesting split-thickness skin grafts and Suprathel^®^ with Omiderm^®^ (Omikron Scientific Ltd., Rehovot, Israel) used on partial-thickness burns.	Suprathel^®^ significantly reduced pain.
Treatment of frostbites—effectiveness of dermal substitute application on frostbite wounds—a case report [64]	Case study	Supporting epidermalization	Suprathel was used in 21 patients including patients with burn wounds, frostbites and with Lyell’s syndrome.	Suprathel seems to be a good dressing that can per- form the function of a temporary epidermal substitute in partial-thickness burns and frostbites, and in Lyell’s syndrome.
Comparison of Suprathel^®^ and allograft skin in the treatment of a severe case of toxic epidermal necrolysis [65]	Case report	Wound healing	A 17-year-old female with a diagnosis of TEN with blistering and epidermal separation affecting 80% of the total body surface area (TBSA).	Significantly reduced exudation with the Suprathel^®^-treated areas, which led to fewer dressing changes, less pain and quicker healing time in contrast to the allograft.
The use of Suprathel^®^ in deep dermal burns: first results of a prospective study [66]	Prospective	Scar formation	18 patients with deep dermal burn wounds underwent tangential excision and the wounds matched deep-partial-thickness areas were covered with 0.1 mm STSGs and Suprathel^®^.	The total Patient Scare Scale of Suprathel areas were similar to the STSG areas and the POSAS showed not to be less for Suprathel compared to STSG. Moreover, patients evaluated Suprathel scars to be less stiff and less different in height.
Split thickness skin graft versus application of the temporary skin substitute suprathel in the treatment of deep dermal hand burns: a retrospective cohort study of scar elasticity and perfusion [67]	Prospective	Scar formation	A case series of 80 patients with deep dermal hand burns was examined.	Analysis gave similar results.
Suprathel^®^-assisted surgical treatment of the hand in a dystrophic epidermolysis bullosa patient [68]	Case report	Wound healing	The use of Suprathel after degloving in a 14-year-old boy with EB undergoing surgery due to hand contractures.	Almost complete epidermalization was observed within one week after surgery
Suprathel^®^ causes less bleeding and scarring than Mepilex^®^ (Mölnlycke Health Care AB, Göteborg, Sweden) Transfer in the treatment of donor sites of split-thickness skin grafts [69]	Comparison	Wound healing, scar formation	22 donor sites were examined, each covered side by side with Suprathel^®^ and Mepilex^®^.	Significantly less pain and bleeding was associated with Suprathel^®^ treatment compared with Mepilex^®^ and Suprathel produced a better scar.
Comparison of wound healing and patient comfort in partial-thickness burn wounds treated with SUPRATHEL and epictehydro wound dressings [70]	Comparison	Wound healing	20 patients aged 18 to 75 years who sustained partial- thickness flame, scald, or contact burns with more than 0.5% of their total body surface area.	Interestingly, both dressings showed similar results and can be used alternatively.
Porcine Xenograft and Epidermal Fully Synthetic Skin Substitutes in the Treatment of Partial-Thickness Burns: A Literature Review [71]	Literature Review	Wound healing	Sixteen Suprathel^®^ and 12 porcine xenograft studies were included.	Suprathel^®^ appears to enable wound healing better than PX and reduces burn wound progression.
Suprathel-acetic acid matrix versus acticoat and aquacel as an antiseptic dressing: an in vitro study [72]	Experimental	Antibacterial	The dressings were put on top of the agar plate with superimposed bacterial cultures from the burn unit.	Suprathel^®^ showed an excellent bactericidal effect superior to that of Acticoat^®^ (Smith & Nephew, Watford, UK) and Aquacel^®^ ( ConvaTec, Princeton, NJ, USA).
Paradigm Shift in Treatment Strategies for Second- Degree Burns Using a Caprolactone Dressing (Suprathel^®^)? A 15-Year Pediatric Burn Center Experience in 2084 Patients [73]	Retrospective	Wound healing	The group of 2084 pediatric patients suffering from mixed superficial and deep dermal second-degree burns who treated caprolactone membranes	Less need for skin grafts (15.69%) and fewer procedures required to be performed under general anesthesia (54.35%) compared to alternative dressing materials.
Suprathel^®^ or Mepilex^®^ Ag for treatment of partial thickness burns in children: A case control study [74]	Retrospective	Wound healing	Assessment of healing time, burn wound infection (BWI), need for surgery, and number of dressing changes in 58 children treated with Suprathel^®^ or Mepilex^®^ Ag.	No significant differences were found in any of the outcomes.
Toxic epidermal necrolysis management with Suprathel^®^ [75].	Case report	Wound healing	4-year-old boy with a diagnosis of TEN.	Acceleration of wound healing and epithelialization was observed
Usability and effectiveness of Suprathel^®^ in partial thickness burns in children [54]	Prospective	Wound healing, scar formation	Evaluation adherence of Suprathel^®^ to the wound bed, re-epithelialization time, grafting, wound colonization and infection, pain, dressing changes, length of hospital stay (LOS), and scar formation.	Suprathel^®^ provides potential pain relief and scarring benefits, but extensive wound debridement is required before dressing is applied.
Role of Suprathel^®^ in Dermal Burns in Children [76]	Retrospective	Wound healing	65 children (25 females, 40 males: mean age 4.9 years, range 04 months to 11 years) with dermal burns were treated with Suprathel.	Suprathel is a useful skin alternative for dealing with deep dermal and mid-dermal burns in children.
Use of a Polylactide-based Copolymer as a Temporary Skin Substitute for a Patient With Moist Desquamation Due to Radiation [77]	Case report	Wound healing	The use of a polylactide-based copolymer for covering the skin defects of a patient with moist desquamation due to radiation.	Aplication of polylactid-based copolymer reduced pain and enabled administration of the full radiation protocol preventing skin from deep radiative injury.

## Data Availability

Not applicable.

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
