# Peer review of "Alloplastic Epidermal Skin Substitute in the Treatment of Burns"

_life, 2023, doi:10.3390/life14010043_

Round 1

Reviewer 1 Report

Comments and Suggestions for Authors

The review topic is too broad. 

There is no significant findings or relevant conclusions that can be obtained from this manuscript.

Therefore, authors need to revise the title and the content to be more focus, for example 'Alloplastic Epidermal skin substitute in the treatment of burn'.

Need to add more latest references.

Reviewer 2 Report

Comments and Suggestions for Authors

Thank you for asking me to review this manuscript. The four authors have reviewed the efficacy of Suprathel in the treatment of burns. They conclude that the dressing is a good option in the treatment of burns, especially in children.

1. The manuscript is lengthy at 14 pages. There would appear considerable scope for precis. This could be readily achieved by the avoidance of repetition between the introduction and discussion. 

2. Much of the introduction describes burn wound characteristics at a basic level. The editor may consider that this could be assumed knowledge of a reader at or above medical student level.

3. Whilst there are 79 references, many of those cited have not been critically reviewed. I note that none of the references cited have been published in Life.

4. Table 1 summarises articles published that have evaluated Suprathel. A number of these are just case reports and should not be evaluated in the same way as those studies that have included 20, 80 or in one study >2000 patients.

5. Why did the authors chose to review just one skin substitute? How did they critically assess the benefits of Suprathel given the lack of objective data on burn depth in the majority of the studies cited?

Comments on the Quality of English Language

This could be improved with a number of odd, idiosyncratic or lay terms scattered throughout the manuscript. For example: "lost skin" in abstract; "death of the organism" and "skin transplant" in the introduction.

Reviewer 3 Report

Comments and Suggestions for Authors

Line 19 abstract the sentence Alloplastic epidermal substitute syntax is incorrect.

Line 58 in the introduction ‘’prevent the infection’’ should read prevent infection.

Line 50 intro ‘’may lead sepsis’’ should read may lead to sepsis.

In the end of the introduction it is missing a statement of the purpose of the review.

Methods   State that 25 articles were found in the databases with the search terms but only 16 studies included in the final article for review.  Can the authors explain why 9 articles were excluded and the rationale for this?

More detailed information should be given on the role of M2 macrophages in the healing response.

References

Finnerty CC, Jeschke MG, Herndon DN, Gamelli R, Gibran N, Klein M, Silver G, Arnoldo B, Remick D, Tompkins RG Investigators of the Inflammation and the Host Response Glue Grant. Temporal cytokine profiles in severely burned patients: a comparison of adults and children. Mol Med. 2008;

Li X, Chaudry IH, Choudhry MA. ERK and not p38 pathway is required for IL-12 restoration of T cell IL-2 and IFN-γ in a rodent model of alcohol intoxication and burn injury1. J Immunol. 2009

A figure demonstrating the role of innate immune cells in the response to healing in burns should be included.

Line 163 the wording ‘’keratinocytes are definitely worth mentioning’’ is inappropriate. Please reword.

Can figure 2 have a scale bar included for reference.

Comments on the Quality of English Language

Please proof read carefully their are some common syntax and grammar errors.

Reviewer 4 Report

Comments and Suggestions for Authors

An interesting review paper on Suprathel.

The language is nice but needs some revision (spelling errors etc).

Lines 36-37, please write the complete division, ie Superficial, Superficial dermal, Deep dermal, Full thickness. Later in the paper please stick to these words. Becomes confusing when sometimes referred to “split-thickness burns”, “deep burns”, etc

Line 41 – eventhough 1-3 weeks may be clinical, the sort of definition of superficial dermal burn is that it heals within 2 weeks.

Line 44 – all wounds heal, even full thickness, just a matter of time. When writing deep dermal heals within 2-9 weeks please stress the downsides of waiting that long. Ie needs surgery.

Line 45 – “Full-thickness burns of the dermis and deep burns….” Stick to the definitions from line 36. A full thickness burn equals the complete epidermis and dermis. Sometimes (previously) one talked of 4th degree burn however that term is obsolete.

Line 50 – burns elicit not only local, but also systemic inflammatory reactions.

Line 52 – I do not agree that hydrosurgery is a less invasive technique. Principal is same as cutting with knife however with water. Necrotic tissue is removed.

Lines 68-69 – the goal of burn treatment is to ensure rapid wound closure!

Line (92-)96-99 – I do not understand this section. First the wound healing process is described followed by stressing that the proliferative phase is crucial (I would argue that all phases are crucial for wound healing!). Then you make a connection between angiogenesis and epidermal coverage which is important for ?partial-thickness burns? (superficial dermal? Deep dermal?) – why would angiogenesis be important in potentially spontaneous healing (reepithelializing) burns? You then go on with dysregulation of the proliferative phase that could turn (the superficial dermal?) burn into a deep chronic wound without surgery and transplantation. I don’t follow?

Line 128 – in the enumeration of dressings providing a moist, optimal environment you add silver dressings. However, silver dressings come in many variations whereof plenty are not providing a moist environment.

Line 135 – “…xenografts, allografts and amniotic membrane”. Amniotic membrane is an allograft! However, I assume that you mean allograft as in donor skin – please clarify this and do not treat amniotic membrane as in a class of its own.

Lines 155-160 – which dressing are you writing about here? Omiderm?

Line 166, and more – I don’t think cultured epithelial autografts should be labelled as “dressing”, in analogy with that STSG are not called dressings.

Figure 1 – “Bliological” is erroneously spelled.

Lines 197-198 – please stick to wording defined on line 36-37. 2a would be superficial dermal, 2b deep dermal, 3 full thickness.

Page 7, table, right column “The total patient scare scale…..” Do you mean POSAS? Patient-Observer scar scale? And how to interpret “…the total Observer Scar Scale showed not to be less for Suprathel or was even significantly higher”?

Comments on the Quality of English Language

Writing is good but some spelling errors and erroneous capital letters.
